# Glucose-Lowering Effects of Imeglimin and Its Possible Beneficial Effects on Diabetic Complications

**DOI:** 10.3390/biology12050726

**Published:** 2023-05-16

**Authors:** Hidekatsu Yanai, Hiroki Adachi, Mariko Hakoshima, Hisayuki Katsuyama

**Affiliations:** Department of Diabetes, Endocrinology and Metabolism, National Center for Global Health and Medicine Kohnodai Hospital, 1-7-1 Kohnodai, Chiba 272-8516, Japan; dadachidm@hospk.ncgm.go.jp (H.A.); d-hakoshima@hospk.ncgm.go.jp (M.H.); d-katsuyama@hospk.ncgm.go.jp (H.K.)

**Keywords:** β-cell, endoplasmic reticulum, glucose-stimulated insulin secretion, imeglimin, mitochondria, endoplasmic reticulum

## Abstract

**Simple Summary:**

Mitochondria are organelles found in the cells of animals. Mitochondria use aerobic respiration to generate adenosine triphosphate. Mitochondrial dysfunction is a prominent pathological feature of type 2 diabetes. Imeglimin is a novel oral hypoglycemic agent with a unique mechanism of action that targets mitochondrial bioenergetics. Imeglimin reduces the production of reactive oxygen species that are harmful to the human body. Additionally, it improves function of mitochondria and the endoplasmic reticulum, which are important in the synthesis, folding, modification, and transport of proteins. Imeglimin enhances glucose-stimulated insulin secretion and inhibits the apoptosis of β-cells in the pancreas by the maintaining function and structure of mitochondria and the endoplasmic reticulum in β-cells. Further, imeglimin inhibits hepatic glucose production and ameliorates insulin sensitivity. These mechanisms improve glucose metabolism in patients with type 2 diabetes. Clinical trials into the effects of imeglimin exhibited a good hypoglycemic efficacy and safety profile in type 2 diabetic patients. Interestingly, imeglimin improved vascular dysfunction in patients with type 2 diabetes. In animals, imeglimin improved cardiac and kidney function, and reduced ischemia-induced brain damage. In addition to its glucose-lowering effects, imeglimin can be a useful therapeutic option for diabetic complications in type 2 diabetic patients.

**Abstract:**

Mitochondrial dysfunction is a prominent pathological feature of type 2 diabetes, which contributes to β-cell mass reduction and insulin resistance. Imeglimin is a novel oral hypoglycemic agent with a unique mechanism of action targeting mitochondrial bioenergetics. Imeglimin reduces reactive oxygen species production, improves mitochondrial function and integrity, and also improves the structure and function of endoplasmic reticulum (ER), changes which enhance glucose-stimulated insulin secretion and inhibit the apoptosis of β-cells, leading to β-cell mass preservation. Further, imeglimin inhibits hepatic glucose production and ameliorates insulin sensitivity. Clinical trials into the effects of imeglimin monotherapy and combination therapy exhibited an excellent hypoglycemic efficacy and safety profile in type 2 diabetic patients. Mitochondrial impairment is closely associated with endothelial dysfunction, which is a very early event in atherosclerosis. Imeglimin improved endothelial dysfunction in patients with type 2 diabetes via both glycemic control-dependent and -independent mechanisms. In experimental animals, imeglimin improved cardiac and kidney function via an improvement in mitochondrial and ER function or/and an improvement in endothelial function. Furthermore, imeglimin reduced ischemia-induced brain damage. In addition to glucose-lowering effects, imeglimin can be a useful therapeutic option for diabetic complications in type 2 diabetic patients.

## 1. Introduction

The production of adenosine triphosphate (ATP) by mitochondria in β-cells is very important for stimulus–secretion coupling, the cascade of molecular events encompassing the initial sensing and transport of glucose to β-cells, and the triggering the exocytosis of insulin [1]. Glucose-stimulated insulin secretion (GSIS) requires increased mitochondrial ATP production up to the metabolic threshold required for insulin release [2,3,4,5,6,7,8,9]. Mitochondrial oxidative phosphorylation (OXPHOS) is crucial to the combination of glucose metabolism with the generation of the ATP required to stimulate insulin secretion. Therefore, an impairment of β-cell mitochondrial ATP production leads to insufficient insulin secretion [3,4,5,6,7,8,9,10,11,12,13]. In patients with type 2 diabetes, reduced ATP generation by the mitochondria and β-cell dysfunction is observed. Abnormal mitochondrial morphology and reduced GSIS are observed in β-cells in patients with type 2 diabetes [7].

The β-cell dysfunction is an early event in the onset of type 2 diabetes. The dysfunction of electron transport chain (ETC) in mitochondria due to excessive reactive oxygen species (ROS) generation may cause β-cell dysfunction. This is because β-cells are highly sensitive to oxidative stress due to their low antioxidant defense mechanism [14,15,16]. Elevated ROS production and inflammation due to mitochondrial dysfunction may exert a significant influence on endothelial cells [17]. Endothelial dysfunction induced by mitochondrial dysfunction causes microangiopathies such as diabetic kidney disease (DKD) and chronic kidney disease (CKD), as well as macroangiopathies such as cardiovascular diseases (CVD) and ischemic stroke in patients with type 2 diabetes. In DKD, glomerular and proximal tubular dysfunction are induced by sustained hyperglycemia and glomerular hyperfiltration [18]. Mitochondrial dysfunction is significantly associated with the progression of DKD, and the pharmacological intervention to treat mitochondrial dysfunction in patients with DKD can be a potential therapeutic option to retard DKD progression [18].

Patients with type 2 diabetes are likely to develop heart failure (HF). HF canbe induced by pathological heart remodeling due to mitochondrial dysfunction [19]. Vice versa, the improvement in mitochondrial dysfunction can improve HF. Mitochondrial dysfunction is one of the mechanisms that initiates the irreversible deterioration of diabetic cardiomyopathy [20]. The induction of mitophagy, which is a process involving the autophagic targeting and clearance of mitochondria destined for removal, can be a promising strategy for diabetic cardiomyopathy [20]. Furthermore, the recovery of mitochondrial function is associated with an improvement in ischemic stroke in animal models [21].

Mitochondrial dysfunction is associated with the development of type 2 diabetes and is also associated with diabetic complications. Imeglimin is a novel agent currently in development to treat type 2 diabetes and has also been shown to normalize glucose tolerance and improve insulin sensitivity by protecting mitochondrial function from oxidative stress [22]. Imeglimin may enhance GSIS and the inhibit apoptosis of pancreatic ß-cells, leading to preserved β-cell mass by maintaining or restoring the functional and structural integrity of mitochondria in β-cells [23]. Imeglimin has the potential to impact the main pathophysiologic components of type 2 diabetes: impaired glucose uptake by muscle tissue, excess hepatic gluconeogenesis, and increased β-cell apoptosis [24]. Therefore, imeglimin use, which improves mitochondrial function, has the possibility to improve not only plasma glucose, but also diabetic complications.

In this study, we discuss the glucose-lowering mechanisms of imeglimin, the ability to decrease HbA1c via imeglimin treatment in clinical trials, the effects of imeglimin on the markers of insulin secretion and insulin resistance and on serum lipids, the safety profile of imeglimin, the effects of imeglimin on endothelial dysfunction, the beneficial effects of imeglimin on the diabetic complications observed in animals, concepts that are currently being demonstrated, and matters requiring elucidated in the future with regard to the abovementioned issues.

## 2. Glucose-Lowering Mechanisms of Imeglimin

### 2.1. An Improvement in Mitochondrial Function

Mitochondria are organelles found in the cells of animals. Mitochondria use aerobic respiration to generate ATP. Imeglimin is a new oral hypoglycemic agent with a unique mechanism for the actions targeting mitochondrial bioenergetics. Mitochondrial dysfunction is a prominent pathological feature of type 2 diabetes, which contributes to worse plasma glucose control via β-cell mass reduction and insulin resistance [25,26,27,28,29,30,31]. Vial et al. characterized the anti-diabetic effects of imeglimin using a high-fat, high-sucrose diet (HFHSD) mice [22]. Imeglimin significantly reduces plasma glucose, and improves insulin sensitivity in HFHSD mice; however, body weight and food intake do not change in such mice. An improvement in glucose metabolism and insulin sensitivity may be induced by an increase in insulin-stimulated protein kinase B phosphorylation in the liver and skeletal muscles. In mitochondria, imeglimin redirects substrate flux towards complex II, inhibits complex I, restores complex III activity, and promotes fatty acid (FA) oxidation. Imeglimin also decreases ROS production and increases mitochondrial DNA. Furthermore, the effects of imeglimin on mitochondrial phospholipid composition may contribute to the improvement in mitochondrial function.

### 2.2. An Improvement in Function of Endoplasmic Reticulum (ER)

ER is a continuous membrane system that plays an essential role in the folding and processing of secretory proteins such as insulin. The excess accumulation of “poorly folded proteins”, which marks the induction of pathogenic ER stress in pancreatic β-cells, largely contributes to the pathogenesis of diabetes [32]. In isolated mouse islets, imeglimin modulates the expression of ER-related various molecules in β-cells under ER stress, and restores protein synthesis in β-cells [33]. Imeglimin significantly decreases apoptotic β-cells and increases β-cell mass in Akita mice. Imeglimin-mediated improvement in structural integrity and homeostasis of ER largely contributes to an enhancement of GSIS [33].

### 2.3. An Inhibition of Hepatic Glucose Production

Imeglimin dose-dependently inhibits hepatic glucose production by increasing mitochondrial redox potential and by decreasing membrane potential in rat hepatocytes [34]. In another study, the authors observed that imeglimin inhibits glucose production in hepatocytes isolated from Wistar rats [35]. Imeglimin markedly inhibits the gluconeogenesis by reducing the phosphoenolpyruvate carboxykinase (PEPCK) and glucose-6-phosphatase (G6Pase) in rat hepatocytes [35].

### 2.4. An Improvement in GSIS

An improvement in mitochondrial dysfunction via imeglimin induces anti-diabetic effects such as the amplification of GSIS and preservation of β-cell mass [33]. Imeglimin improves the markers for β-cell function, such as proinsulin/insulin in humans [36].

Nicotinamide phosphoribosyltransferase (NAMPT) regulates the intracellular nicotinamide adenine dinucleotide (NAD) pool. NAD is an essential coenzyme which is necessary for cellular redox reactions [37]. In metabolic disorders, the levels of NAD are decreased. NAMPT can change the pathogenesis of obesity and type 2 diabetes by modifying oxidative stress response, apoptosis, lipid and glucose metabolism, inflammation, and insulin resistance [37]. Imeglimin acutely and directly amplifies GSIS in rodents with type 2 diabetes. The elevation of cellular NAD via the salvage pathway and the induction of NAMPT with the enhancement of glucose-induced ATP levels may amplify GSIS [38]. The conversion of NAD into a cyclic ADP ribose is required for GSIS, suggesting a significant association between elevated NAD and augmented glucose-induced Ca^2+^ mobilization which induces the exocytosis of insulin granule [38]. An imeglimin-mediated increase in intracellular calcium may enhance insulin secretion [38]. Further, imeglimin reduces β-cell apoptosis by lowering the glucotoxicity via a mitochondrial-improvement-dependent mechanism [39]. Imeglimin increases β-cell mass by inhibiting the permeability transition pores (PTP) of mitochondria [39]. Imeglimin increases insulin secretion in a glucose-dependent manner, and also exerts beneficial effects on β-cell mitochondrial integrity in type 2 diabetic mice [40]. The improvement in β-cell mitochondria may facilitate ATP production, enhancing the synthesis and secretion of insulin.

### 2.5. An Improvement in β-Cell Function

Imeglimin has a favorable effect on the preservation of the number of insulin granules, the recovery of mitochondrial structure, and the reduction in apoptosis [40]. How could imeglimin improve pancreatic β-cell function? The reduced expression of apoptosis- and inflammation-associated factors such as inflammatory cytokines may prevent β-cell apoptosis. A decrease in oxidative stress by imeglimin may also lead to reduceed β-cell apoptotic cell death and to improved β-cell function. The decrease in β-cell death by imeglimin may be closely linked with the amelioration of β-cell function. In the situation with the induction of apoptotic β-cell death, it is hard for β-cells to preserve the synthesis and secretion of insulin. Although further studies should be performed, the imeglimin-mediated prevention of apoptotic β-cell death may improve β-cell function. Briefly, the improvement in β-cell mitochondrial structure is likely to facilitate ATP production, enhancing β-cell function. Furthermore, imeglimin-mediated improvement in structural integrity and homeostasis of ER may largely contribute to an improvement in β-cell function [33].

### 2.6. An Enhancement of Glucose Uptake by the Skeletal Muscles

Skeletal muscle is the major tissue involved in insulin-mediated glucose disposal. The decreased uptake of glucose by muscle due to insulin resistance is an important factor in the development of type 2 diabetes. Acute administration of imeglimin significantly stimulates glucose uptake by muscle cells in a dose-dependent manner [35]. Muscle glucose uptake is reduced in the streptozotocin-treated rats compared to the control rats. Chronic administration of imeglimin (45 days) increases glucose uptake by such muscles. An improvement in mitochondrial dysfunction by imeglimin induces an improvement in insulin signaling in skeletal muscle [35]. An improvement in insulin resistance by imeglimin is expected to increase glucose transporter 4 (GLUT4) expression and improve GLUT4 function in muscle, but this has not yet been proven.

### 2.7. An Improvement in Oxidative Stress and Insulin Resistance

Oxidative stress has a crucial role in the pathogenesis of diabetes and its complications [41,42]. Oxidative stress can induce insulin resistance by impairing various insulin signaling [42]. Imeglimin has antioxidative properties which enable it to ameliorate free radical generation and readjust the redox state [22]. Imeglimin reduces oxidative stress by suppressing the mitochondrial free radical generation, which improves glucose homeostasis [23].

Vial et al. show that imeglimin improves insulin sensitivity in high-fat diet mice [22]. Pacini et al. demonstrate that imeglimin improves insulin sensitivity in β-cells in type 2 diabetic patients [43]. They suggest that imeglimin can increase peripheral insulin sensitivity in patients with diabetes [43]; however, the mechanism for this remains largely unknown.

The summary of glucose-lowering mechanisms of imeglimin was shown in Figure 1. Imeglimin improves mitochondrial function, which ameliorates insulin sensitivity in liver and skeletal muscle. As a result, FA oxidation increases in liver and skeletal muscle, hepatic glucose production decreases, and glucose uptake increases in skeletal muscle. Furthermore, imgelimin improves mitochondrial and ER function, which increases GSIS and β-cell mass. Such improvements in insulin sensitivity in metabolically active organs may improve systemic insulin resistance, which may change the insulin resistance milieu in adipose tissue in patients with type 2 diabetes, such as via an increase in adiponectin and a decrease in inflammatory cytokines.

## 3. Glucose-Lowering Effects of Imeglimin in Clinical Trials

### 3.1. Effects of Imeglimin on HbA1c in Clinical Trials

The changes induced in HbA1c from baseline by imeglimin monotherapy and imeglimin combination therapy are shown in Table 1 [44,45,46,47,48,49,50].

To assess the efficacy of the 24-week imeglimin monotherapy vs. placebo in Japanese type 2 diabetic patients, a randomized, double-blind, placebo-controlled, dose-ranging clinical trial was performed. Patients were randomly assigned to receive imeglimin 500, 1000 or 1500 mg, or a placebo treatment twice-daily over a 24-week period [45]. At week 24, imeglimin significantly decreased HbA1c as compared with placebo: imeglimin (1000 mg/day), −0.52% (95% CI, −0.77% to −0.27%); imeglimin (2000 mg/day), −0.94% (95% CI, −1.19% to −0.68%); imeglimin (3000 mg/day), −1.0% (95% CI, −1.26% to −0.75%; *p* < 0.0001 for all). The difference in efficacy between those treated with 3000 mg/day and those with 2000 mg/day was small and gastrointestinal symptoms were raised; a daily dose of 2000 was used for the phase 3 studies.

As seen in Table 1, a daily 1000 mg dose of imeglimin was insufficient to improve HbA1c in the short term (8 weeks) and relatively long term (24 weeks) [44,45]. As seen by comparing changes in HbA1c at 8 and 24 weeks after the start of the administration of imeglimin (3000 mg/day), it takes a long time over 8 weeks to obtain a sufficient reduction in HbA1c via imeglimin [44,45]. A daily 2000 mg dose of imeglimin showed the same degree of decrease in HbA1c at 24 and 52 weeks after the start of imeglimin [45,47], suggesting that the imeglimin monotherapy shows a sustained hypoglycemic effect.

The addition of imeglimin (3000 mg/day) to metformin or sitagliptin decreased HbA1c by 0.65% and by 0.6%, respectively, in patients with type 2 diabetes after 12 weeks [48,49]. The efficacy of the combination of imeglimin with metformin was confirmed in Japanese patients with type 2 diabetes. This combination therapy reduced HbA1c by 0.67% from the baseline after 52 weeks of treatment [47]. This suggests the complementary action of two drugs: metformin primarily suppresses hepatic excessive glucose production, whereas imeglimin influences both insulin sensitivity and glucose-dependent insulin secretion.

Imeglimin, in combination with dipeptidyl peptidase-4 inhibitors (DPP4i) such as sitagliptin, induced a significant HbA1c decrease of 0.92% from baseline after 52 weeks [47]. This suggests that their mechanisms of action are different and that their effects on glucose control complement one another to induce more potent efficacy when combined, although both DPP4i and imeglimin increase insulin secretion in response to glucose.

The efficacy and safety of the 52-week imeglimin treatment combined with insulin in Japanese patients with type 2 diabetes was evaluated by several authors [50]. Compared with placebo, imeglimin reduced HbA1c by 0.60% after 16 weeks. This decrease was sustained up to 52 weeks by 0.64% versus baseline. The number of patients experiencing hypoglycemia was similar in the two treatment groups.

### 3.2. Effects of Imeglimin on Fating Plasma Glucose (FPG), the Markers for Insulin Secretion and Insulin Resistance and Serum Lipids in Clinical Trials

The effects of imeglimin on FPG, the markers for insulin secretion and insulin resistance and serum lipids in clinical trials, were shown in Table 2.

In all studies using over 2000 mg of imeglimin daily, FPG was significantly decreased. FPG is largely determined by hepatic gluconeogenesis during the nighttime fasting period. A decrease in FPG may be induced by the inhibition of hepatic gluconeogenesis by imeglimin.

The proinsulin/insulin ratio can be the predictor of insulin resistance and β-cell dysfunction [51]. The proinsulin-to-C-peptide ratio may indicate chronic β-cell stimulation with β-cell loss as well as the secretion of immature insulin granules [52]. Increased proinsulin secretion relative to fully processed insulin may reflect β-cell ER dysfunction [53]. Moreover, an elevated proinsulin-to-C-peptide ratio has been observed in patients with prediabetes, as well as type 1 and type 2 diabetes [52,54,55]. The homeostatic model assessment (HOMA) HOMA of β-cell function (HOMA-B) index is a marker for β-cell function [56]. HOMA-B estimates of β-cell function correlate well with estimates using continuous infusion glucose model assessment and the hyperglycemic clamp [57].

A reduction in the proinsulin/insulin ratio and proinsulin-to-C-peptide ratio was observed in three clinical trials [45,46,48], just as an increase in HOMA-B was observed in all trials which measured HOMA-B, suggesting an improvement in β-cell function and GSIS by imeglimin. Imeglimin increased insulin sensitivity in high-fat diet mice [22]. However, the homeostasis model assessment of insulin resistance (HOMA-IR) did not show a significant change in the all trials that measured HOMA-IR. Furthermore, serum lipids did not show any significant changes in clinical trials.

### 3.3. Stimulatory Effect of Imeglimin on Incretin Secretion

Imeglimin in combination with DPP4i produced a clinically meaningful HbA1c decrease of 0.92% from baseline values after 52 weeks [47]. The reduction in HbA1c with this combination is significantly greater than the reduction in HbA1c with the combination of imeglimin and sodium glucose cotransporter inhibitors (−0.57%), imeglimin and biguanide (−0.67%), or imeglimin and sulfonyl urea (−0.56%) [47]. Very recently, scholars studied whether incretin hormones contribute to the pharmacological actions of imeglimin in mice [58]. Imeglimin induced an increase in plasma glucagon-like peptide-1 (GLP-1) levels that may contribute at least in part to its stimulatory effect on insulin secretion [58].

## 4. A Safety Profile of Imeglimin

### 4.1. A Safety Profile Obtained from Clinical Trials Which Studied Imeglimin Monotherapy and Combination Therapy

Imeglimin shows a superior safety profile as compared with metformin in type 2 diabetes patients [44]. In the phase 3 trial in Japanese type 2 diabetic patients, 44.3% of patients reported ≥1 adverse events in the imeglimin group, versus 44.9% of patients in the placebo group [46].

The addition of imeglimin to metformin was generally well tolerated, with a comparable safety profile to the combination of placebo with metformin in patients with type 2 diabetes [48]. In the add-on therapy to sitagliptin, imeglimin was well tolerated, with a safety profile similar to that of the placebo and with no serious adverse events [49]. The administration of imeglimin with metformin or sitagliptin did not induce clinically meaningful changes in the systemic exposure to metformin or sitagliptin [59]. In the trial that evaluated the efficacy and safety of imeglimin as a combination therapy with insulin in Japanese type 2 diabetic patients, the incidence of adverse events, serious adverse events, and hypoglycemia was similar in the imeglimin and placebo groups [50]. In patients receiving imeglimin, all hypoglycemic events were mild; no episodes required assistance [50].

### 4.2. Reported Treatment Emergent Adverse Events (TEAEs)

According to the study evaluating the long-term (52 weeks) safety of imeglimin, there were no serious drug-related TEAEs [47]. The TEAEs occurring in more than 5% of patients with imeglimin monotherapy included nasopharyngitis, pharyngitis and nausea [47]. The TEAEs occurring in 3–5% of patients with imeglimin monotherapy included diarrhea, constipation, hypoglycemia and hyperglycemia [47].

### 4.3. Electrophysiological Effects of Imeglimin on Cardiac Repolarization

The effect of therapeutic and supratherapeutic doses of imeglimin on electrophysiological changes was evaluated [60]. Healthy volunteers were given doses of 2250 mg and 6000 mg of imeglimin. Such therapeutic and supratherapeutic doses of imeglimin did not induce the prolongation of QT interval.

### 4.4. The Drug–Drug Interaction between Imeglimin and an Inhibitor of Human Multidrug and Toxic Extrusion Transporters (MATE) and Organic Cation Transporters (OCT)

In vitro, imeglimin is a substrate of human MATE1, MATE2-K, OCT1, and OCT2. The potential drug–drug interaction of imeglimin with cimetidine, reference inhibitor of such transporters, was evaluated [61]. Clinically significant drug–drug interactions were not observed.

### 4.5. The Effect of Hepatic Impairment on the Pharmacokinetics (PK) of Imeglimin

Imeglimin is mainly excreted in its unchanged form by the kidney; however, it is a substrate of OCTs, which are also expressed in the liver. The effect of hepatic impairment on the PK of imeglimin was assessed [62]. Imeglimin was safe and well tolerated in subjects with moderate hepatic impairment.

### 4.6. The Effect of Renal Impairment on the PK of Imeglimin

The PK and safety of using imeglimin in patients with renal dysfunction was evaluated [63]. Plasma concentration (Cmax) and area-under-the-curve (AUC) values were higher in patients with impaired renal function than in patients with normal renal function. Renal clearance of imeglimin decreased with the reduction in renal function. The Cmax and AUC values were higher after multiple doses in patients with renal impairment than patients with normal renal function. No adverse events were observed. Dose adjustment is required in patients with moderate and severe renal impairment and has an estimated glomerular filtration rate (eGFR) of 15–45 mL/min/1.73 m^2^.

Another study was performed to define the PK characteristics of imeglimin and to determine the optimal dose for Japanese patients with type 2 diabetes and CKD [64]. The eGFR, body weight, and age were important determinants of imeglimin clearance. The administered dose was the main determinant of the absorbed imeglimin. The administration of 500 mg twice daily was recommended for patients with eGFR 15–45 mL/min/1.73 m^2^. A longer dosing interval for dosing would be required for patients with eGFR less than 15 mL/min/1.73 m^2^.

### 4.7. Reduced Lactic Acidosis Risk with Imeglimin as Compared with Metformin

Metformin, biguanide, is widely used for patients with type 2 diabetes; its use is associated with the development of lactic acidosis, in particular, in patients with renal failure and major surgery. Imeglimin has a similar chemical moiety to metformin and has an effect on mitochondrial complex I activity, which is a possible mechanism for the development of lactic acidosis by metformin. In a dog model of major surgery, the high dose of metformin or imeglimin was acutely administered, but only metformin developed lethal lactic acidosis [65]. A high dose of metformin or imeglimin was administered to rats with renal insufficiency, but only metformin developed fatal lactic acidosis [65]. In models of both dog and rats, plasma concentrations of metformin and imeglimin were similar. Greater inhibition of mitochondrial complex I was observed with metformin, whereas such inhibition with imeglimin was slight. Only metformin showed an inhibitory effect on the mitochondrial glycerol-3-phosphate dehydrogenase (mGPDH). Metformin was reported to inhibit mGPDH, resulting in the decrease in entry of glycerol into gluconeogenic flux, disrupting the glycerophosphate shuttle and inducing accumulation of cytosolic NADH which is closely associated with the development of lactic acidosis [66,67].

## 5. Effects of Imeglimin on Endothelial Dysfunction, Heart, Kidney and Brain

### 5.1. Effects of Imeglimin on Endothelial Dysfunction

Endothelial dysfunction is a crucial risk factor for CVD in diabetic patients. To clarify whether imeglimin improves endothelial function, imeglimin was administered to type 2 diabetic patients for 3 months, and flow-mediated vasodilation (FMD) was measured [68]. Imeglimin significantly improved HbA1c, FPG and 2 h postprandial glucose. Both patients with and without a decrease in postprandial glucose showed an improvement in postprandial FMD, suggesting that imeglimin might improve endothelial function in both glycemic control-dependent and -independent mechanisms.

Detaille et al. studied the protective effects of imeglimin on the hyperglycemia-induced death of human endothelial cells (HMEC-1) [69]. HMEC-1 were incubated in oxidative stress environments, which induced the opening of mitochondrial permeability transition pore (PTP), the release of cytochrome c release, and cell death. Such events were prevented by imeglimin treatment without any effect on oxygen consumption. Imeglimin dramatically reduced ROS production. This suggests that imeglimin prevents hyperglycemia-induced cell death through the inhibition of the opening of PTP without the inhibition of mitochondrial respiration.

Imeglimin, a glucose-lowering agent targeting mitochondrial bioenergetics, decreases ROS overproduction and improves glucose homeostasis. Lachaux et al. investigated whether such properties are associated with protective effects on metabolic syndrome-related vascular dysfunction in rats [70]. Imeglimin restored acetylcholine-mediated coronary artery relaxation and mesenteric artery FMD, suggesting an improvement in endothelial dysfunction by imeglimin. Imeglimin immediately countered insulin resistance-related vascular dysfunction by reducing oxidative stress and by increasing NO bioavailability.

### 5.2. Effects of Imeglimin on Left Ventricular (LV) Function

In the study by Lachaux et al. [70], imeglimin administration reduced LV mitochondrial ROS production and improved LV function after one hour in rats. Ninety-day imeglimin treatment reduced LV fibrosis. Imeglimin immediately improved cardiac diastolic dysfunction by reducing oxidative stress, increasing NO bioavailability and improving myocardial perfusion and by using a 90-day-treatment-improved myocardial structure.

Heart failure with preserved ejection fraction (HFpEF), which is HF with a LV ejection fraction (LVEF) >50%, has emerged as a crucial medical problem in type 2 diabetic patients [71,72]. HFpEF patients with metabolic risk factors, including obesity and diabetes, show endothelial dysfunction, cardiac hypertrophy and fibrosis, and myocardial excessive fat accumulation due to increased inflammation and oxidative stress [73,74], which induce a development of diastolic dysfunction. Kitakata et al. investigated the effect of imeglimin on the pathogenesis of HFpEF [75]. They made a murine HFpEF model by a high-fat diet (HFD) and the nitric oxide synthase (NOS) inhibitor N[w]-nitro-l-arginine methyl ester (l-NAME). Mice treated with a HFD (metabolic stress) and the NOS inhibitor l-NAME (mechanical stress) developed diastolic dysfunction and nary congestion with a preserved EF [74]. The development of HFpEF in such a murine model was associated with the dysfunction of the unfolded protein response (UPR) because of the enhancement of nitrosative stress associated with the enhanced NO production by inducible nitric oxide synthase (iNOS). Recently, the enhancement of endothelial barrier function by mild activation of the UPR has been suggested, which is closely associated with thr suppression of abnormal increases of ER stress [76]. An increase in ROS activity upregulates iNOS expression in vitro in human coronary artery endothelial cells (HCAECs) grown in culture, and also in vivo in animals [77]. Imeglimin ameliorated the HFpEF phenotype and cardiac steatosis in a murine HFpEF model by suppressing the expression of iNOS and normalizing the UPR [75].

### 5.3. Effects of Imeglimin on Kidney Function and Structure

In the study by Lachaux et al. [70], the 90-day imeglimin treatment reduced albuminuria, with no reduction in serum creatinine or urinary volume in Zucker rats. The 90 day-imeglimin treatment significantly reduced interstitial fibrosis; while the glomerular injury score and interstitial inflammation were also reduced, the reduction did not reach statistical significance.

### 5.4. Effects of Imeglimin on Ischemia-Induced Brain Damage

The effects of imeglimin on ischemia-induced brain damage, induced in rats by the occlusion of the cerebral artery, was investigated [78]. The treatment with imeglimin significantly reduced the size of cerebral infarction, cerebral edema, and the neurological defects of ischemia. Furthermore, imeglimin protected against ischemia-induced neuronal loss, microglial proliferation and activation, and increased the number of astrocytes and cells that produce anti-inflammatory cytokines such as interleukin-10. Further, imeglimin acutely prevented the opening of PTP in cultured neurons and astrocytes.

The summary of Effects of imeglimin on endothelial function, heart, kidney and brain is shown in Figure 2. Imeglimin improved endothelial function by improving mitochondrial and ER function. Imeglimin improved LV and kidney function by an improvement in mitochondrial and ER function or/and an improvement in endothelial function. Furthermore, imeglimin reduced ischemia-induced brain damage in an animal model.

## 6. Conclusions

The matters currently being demonstrated, the matters that should be elucidated in the future about the mechanisms for plasma glucose-lowering of imeglimin, the effects of imeglimin on glucose and lipid metabolism, and the possible beneficial effects of imeglimin on diabetic complication are shown in Figure 3. Imeglimin improved mitochondrial and ER function; this ameliorated β-cell function, which increased GSIS. Imgelimin inhibited hepatic glucose production and increased muscle glucose uptake in rats. An improvement in insulin resistance was observed in mice; however, an improvement in HOMA-IR as the index of insulin resistance was not observed in randomized controlled trials (RCTs), which should be studied by using real-world data (RWD). A decrease in HbA1c and FPG was observed in most RCTs; however, the effects of imeglimin on glycemic excursion, including postprandial plasma glucose, have never been studied. This issue should be evaluated in the future. Any significant change in serum lipids after the start of imeglimin were not observed in RCTs, which should be re-confirmed using the RWD. Further, the effect of imeglimin on other atherogenic lipoproteins such as remnant, non-HDL-C and oxidized LDL should be studied in the future. Only one human study demonstrated an improvement in FMD by imeglimin; however, further studies to measure FMD, intima-media thickness (IMT) and pulse wave velocity (PWV) as the makers for endothelial dysfunction should be performed. An improvement in LV function by imeglimin was observed in a murine model with HF, which should be studied in humans. Reduced albuminuria, glomerular injury and interstitial inflammation were observed in rats. Changes in eGFR and urinary albumin after the start of imeglimin induction should be evaluated in the future. Currently, we require anti-diabetic drugs with favorable effects on major adverse cardiovascular events (MACE) and renal outcomes, in addition to exhbiting hypoglycemic effects. The effect of imeglimin on such clinical outcomes should also be examined in the future.

## Figures and Tables

**Figure 1 biology-12-00726-f001:**
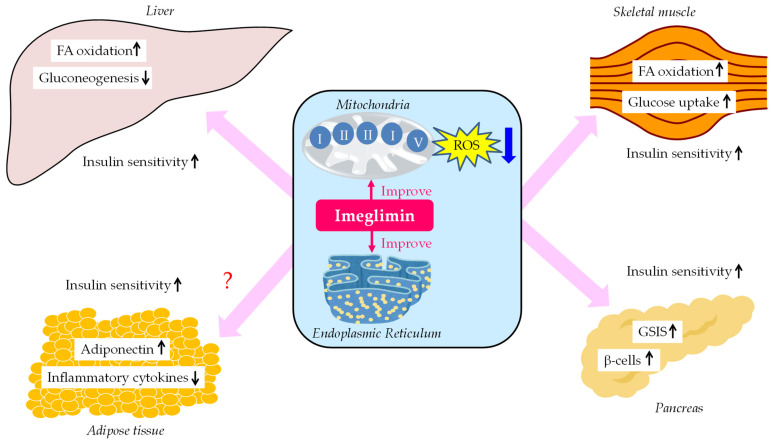
The summary of glucose-lowering mechanisms of imeglimin. FA, fatty acid; GSIS, glucose-stimulated insulin secretion; ROS, reactive oxygen species.

**Figure 2 biology-12-00726-f002:**
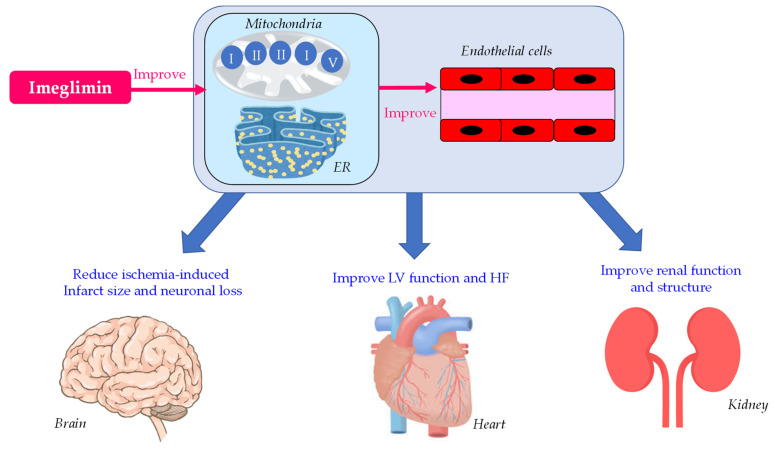
Summary of effects of imeglimin on endothelial function, heart, kidney and brain. ER, endoplasmic reticulum; HF, heart failure; LV, left ventricle.

**Figure 3 biology-12-00726-f003:**
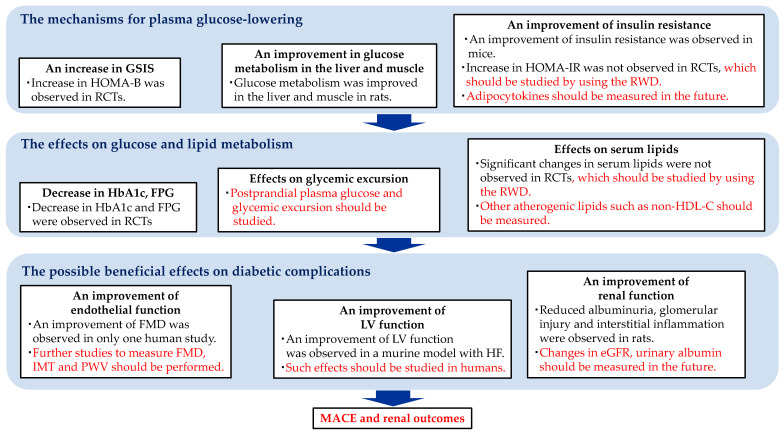
What are currently demonstrated and what should be elucidated in the future about the mechanisms of plasma glucose-lowering of imeglimin, the effects of imeglimin on glucose and lipid metabolism and the possible beneficial effects of imeglimin on diabetic complication. The description in black letters indicates what is currently being demonstrated, and the description in red letters indicate what should be elucidated in the future. eGFR, estimated glomerular filtration rate; GSIS, glucose-stimulated insulin secretion; FMD, flow-mediated vasodilation; FPG, fasting plasma glucose; HF, heart failure; HOMA-B, homeostatic model assessment of β-cell function; HOMA-IR, homeostasis model assessment-insulin resistance; IMT, intima-media thickness; LV, left ventricular; MACE, major adverse cardiovascular events; non-HDL-C, non-high-density lipoprotein cholesterol; PWV, pulse wave velocity; RCTs, randomized controlled trials; RWD, real-world data.

**Table 1 biology-12-00726-t001:** The changes in HbA1c from baseline by imeglimin monotherapy and combination therapy in clinical trials.

Monotherapy			
Daily Dose (mg)	Duration (Weeks)	Change in HbA1c (%)	References
1000	8	+0.38	[44]
3000	8	−0.18	[44]
1000	24	−0.09	[45]
2000	24	−0.51	[45]
2000	24	−0.72	[46]
3000	24	−0.57	[45]
2000	52	−0.46	[47]
Combination therapy			
3000 + BG	12	−0.65	[48]
2000 + BG	52	−0.67	[47]
3000 + DPP4i	12	−0.6	[49]
2000 + DPP4i	52	−0.92	[47]
2000 + insulin	16	−0.63	[50]

BG, biguanide; DPP4i, dipeptidyl peptidase 4 inhibitors.

**Table 2 biology-12-00726-t002:** Effects of imeglimin on fasting plasma glucose, the markers for insulin secretion and insulin resistance and serum lipids in clinical trials.

	Monotherapy						Combination Therapy		
Daily Dose (mg)	1000	3000	1000	2000	2000	3000	3000	3000	2000
Duration (weeks)	8	8	24	24	24	24	12	12	16
							Metformin	Sitagliptin	Insulin
References	[44]	[44]	[45]	[46]	[45]	[45]	[48]	[49]	[50]
FPG	↑	↓	→	↓	↓	↓	↓	↓	↓
Proinsulin/insulin				↓			↓	→	
Proinsulin/C-peptide			→	↓	→	↓			
HOMA-B			↑	↑	↑	↑			
HOMA-IR			→	→	→	→		→	
LDL-C			→		→	→			→
HDL-C			→		→	→			→
Triglyceride			→		→	→		→	→

FPG, fasting plasma glucose; HDL-C, high-density lipoprotein cholesterol; HOMA-B, homeostatic model assessment of β-cell function; HOMA-IR, homeostasis model assessment-insulin resistance; LDL-C, low-density lipoprotein cholesterol.

## Data Availability

This is systematic review article and do not include the original data, and we showed all papers we systematically reviewed in the section of “References”.

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
