# Peer review of "Glucose-Lowering Effects of Imeglimin and Its Possible Beneficial Effects on Diabetic Complications"

_biology, 2023, doi:10.3390/biology12050726_

Round 1

Reviewer 1 Report

The review manuscript have discussed about "Glucose-lowering and Beyond Glucose-lowering Effects of A Novel Anti-diabetic Drug with Action Targeting Mitochondrial Bioenergetics, Imeglimin: Its Effects on Endothelial Cells, Heart, Kidney and Brain".

Based on the scientific content of the review is not provide the sufficient new knowledge to the readers among the scientific society. Therefore I would suggest authors to re-write the whole manuscript with the focus of molecular mechanism of Anti-diabetic Drug.

There are few major comments need to be addressed before accepted for publication

1. Title of the manuscript is not reflecting the content of the manuscript.

2. Introduction part seems very limited information about imeglimin and it’s not convincing concept of the review manuscript.

3. Objective of the manuscript not clear so it has to be addressed properly and provide the adequate details with recent citations.

4. Conclusion part is not completed one so it must be revised with the future directions along with flow chart.

5. Figure 2, must be converted to Table since the present form very difficult to understand.

6. I would suggest authors to add few more recent citations about the present review.

Author Response

Dear Reviewer

Thank you very much for your very reasonable and wonderful suggestions.

Thanks to your suggestions, I was able to realize my mistake and made an improved review article.

  1. According to the suggestion, “Title of the manuscript is not reflecting the content of the manuscript.”

We changed the tile

From

Glucose-lowering and Beyond Glucose-lowering Effects of A Novel Anti-diabetic Drug with Action Targeting Mitochondrial Bioenergetics, Imeglimin: Its Effects on Endothelial Cells, Heart, Kidney and Brain

To

Glucose-lowering Effects of Imeglimin and Its Possible Beneficial Effects on Diabetic Complications

  1. According to the suggestion “Introduction part seems very limited information about imeglimin and it’s not convincing concept of the review manuscript.”

We significantly changed the section of “Introduction” as the followings by citing more 21 new references.

The production of adenosine triphosphate (ATP) by mitochondria in β-cells is very important for the stimulus-secretion coupling, the cascade of molecular events en-compassing the initial sensing and transport of glucose to β-cells, and the triggering the exocytosis of insulin [1]. The glucose-stimulated insulin secretion (GSIS) requires increased mitochondrial ATP production up to the metabolic threshold required for insulin release [2-9]. Mitochondrial oxidative phosphorylation (OXPHOS) is crucial to combine glucose metabolism with the generation of ATP required to stimulate insulin secretion. Therefore, an impairment of β-cell mitochondrial ATP production leads to insufficient insulin secretion [3-13]. In patients with type 2 diabetes, reduced ATP gen-eration by the mitochondria and β-cell dysfunction were observed. Abnormal mito-chondrial morphology and reduced GSIS were observed in β-cells in patients with type 2 diabetes [14].

 The β-cell dysfunction is an early event in the onset of type 2 diabetes. The dys-function of electron transport chain (ETC) in mitochondria due to excessive reactive oxygen species (ROS) generation may cause β-cell dysfunction, because -cells are highly sensitive to oxidative stress due to their low antioxidant defense mechanism [15-17]. Elevated ROS production and inflammation due to mitochondrial dysfunction may give a significant influence on endothelial cells [18]. Endothelial dysfunction in-duced by mitochondrial dysfunction causes microangiopathy such as diabetic kidney disease (DKD) and chronic kidney disease (CKD), and macroangiopathy such as car-diovascular diseases (CVD) and ischemic stroke in patients with type 2 diabetes. In DKD, glomerular and proximal tubular dysfunction are induced by sustained hyper-glycemia and glomerular hyperfiltration [19]. Mitochondrial dysfunction was signifi-cantly associated with the progression of DKD, and the pharmacological intervention for mitochondrial dysfunction in patients with DKD can be a potential therapeutic op-tion to retard DKD progression [19].

The patients with type 2 diabetes are likely to develop heart failure (HF). HF could be induced by pathological heart remodeling due to mitochondrial dysfunction [20]. Vice versa, the improvement of mitochondrial dysfunction can improve HF. Mito-chondrial dysfunction is one of the mechanisms that initiates the irreversible deterio-ration of diabetic cardiomyopathy [21]. The induction of mitophagy, which is the pro-cess involving the autophagic targeting and clearance of mitochondria destined for removal, can be a promising strategy for diabetic cardiomyopathy [21]. Furthermore, the recovery of mitochondria function was associated with an improvement of is-chemic stroke in animal models [22].

Mitochondrial dysfunction is associated with the development of type 2 diabetes and is also associated with diabetic complication. Imeglimin is a novel agent currently in development to treat type 2 diabetes, and was also shown to normalize glucose tol-erance and improve insulin sensitivity by protecting mitochondrial function from oxi-dative stress [23]. Imeglimin may enhance GSIS and inhibit apoptosis of pancreatic ß-cells leading to preserved β-cell mass by maintaining or restoring the functional and structural integrity of the mitochondria in β-cells [24]. Imeglimin has the potential to impact the main pathophysiologic components of type 2 diabetes: impaired glucose uptake by muscle tissue, excess hepatic gluconeogenesis, and increased -cell apoptosis [25]. Therefore, imeglimin which improves mitochondrial function has a possibility to improve not only plasma glucose, but also diabetic complications.

  1. According to the suggestion “Objective of the manuscript not clear so it has to be addressed properly and provide the adequate details with recent citations.”

We significantly changed the section of “Introduction” citing more 21 new references and added the following sentences at the end of “Introduction” to clear the objective of our review article.

Here, we discuss on the glucose-lowering mechanisms of imeglimin, the potency to decrease HbA1c by imeglimin in clinical trials, effects of imeglimin on the markers for insulin secretion and insulin resistance and on serum lipids, the safety profile of imeglimin, the effect of imeglimin on endothelial dysfunction, and beneficial effects of imeglimin on diabetic complication which were observed in animals, and what are currently demonstrated and what should be elucidated in the future about the above mentioned issues.

  1. According to the suggestion “Conclusion part is not completed one so it must be revised with the future directions along with flow chart.”

We changed as the following sentences and made Figure 3 showing the future directions.

What are currently demonstrated and what should be elucidated in the future about the mechanisms for plasma glucose-lowering of imeglimin, the effects of imeglimin on glucose and lipid metabolism and the possible beneficial effects of imeglimin on diabetic complication were shown in Figure 3. Imeglimin improved mi-tochondrial and ER function which ameliorated -cell function which increased GSIS. Imgelimin inhibited hepatic glucose production and increased muscle glucose uptake in rats. An improvement of insulin resistance was observed in mice, however, an im-provement of HOMA-IR as the index of insulin resistance was not observed in ran-domized controlled trials (RCTs), which should be studied by using the real-world data (RWD). A decrease in HbA1c and FPG was observed in most of RCTs, however, effect of imeglimin on glycemic excursion including postprandial plasma glucose have not ever been studied, which should be evaluated in the future. Any significant change in serum lipids after the start of imeglimin were not observed in RCTs, which should be re-confirmed by using the RWD. Further, the effect of imeglimin on other atherogenic lipoproteins such as remnant, non-HDL-C and oxidized LDL should be studied in the future. Only one human study demonstrated an improvement of FMD by imeglimin, however, further studies to measure FMD, intima-media thickness (IMT) and pulse wave velocity (PWV) as the makers for endothelial dysfunction should be performed. An improvement of LV function by imeglimin was observed in a murine model with HF, which should be studied in humans. Reduced albuminuria, glomerular injury and interstitial inflammation were observed in rats. Changes in eGFR and urinary albumin after the start of imeglimin should be evaluated in the future. Currently, anti-diabetic drugs are required to have favorable effects on major adverse cardiovascular events (MACE) and renal outcomes in addition to hypoglycemic effects. The effect of imeglimin on such clinical outcomes should be also examined in the future.

  1. According to the suggestion “Figure 2, must be converted to Table since the present form very difficult to understand.”

We changed from Figure 2 to Table 1.

  1. I would suggest authors to add few more recent citations about the present review.

We deleted 13 references on SGLT2 inhibitors and GLP-1 receptor analogue and added total 22 references on mitochondrial dysfunction and imeglimin.

Furthermore, we added the following sentences by citing a new reference.

3.3. Stimulatory effect of imeglimin on incretin secretion

Imeglimin in combination with DPP4i produced a clinically meaningful HbA1c decrease of 0.92% from baseline after 52 weeks [48]. The reduction in HbA1c with this combination is significantly greater than the reduction in HbA1c with the combination of imeglimin and sodium glucose cotransporter inhibitors (-0.57%) or imeglimin and biguanide (-0.67%) or imeglimin and sulfonyl urea (-0.56%) [48]. Very recently, it was studied whether incretin hormones might contribute to the pharmacological actions of imeglimin in mice [59]. Imeglimin induced an increase in plasma glucagon-like pep-tide-1 (GLP-1) levels which may contribute at least in part to its stimulatory effect on insulin secretion [59].

Reviewer 2 Report

The review report is attached herewith. 

Author Response

Dear Reviewer

Thank you very much for your very reasonable and wonderful suggestions.

Thanks to your suggestions, I was able to realize my mistake and made an improved review article.

  1. According to the suggestion “The authors reviewed the “Glucose-lowering and Beyond Glucose-lowering Effects of A 2 Novel Anti-diabetic Drug with Action Targeting Mitochondrial 3 Bioenergetics, Imeglimin: Its Effects on Endothelial Cells, 4 Heart, Kidney and Brain. In general, the manuscript brings relevant information. However, to recommend the work for publication, the authors must carry out revisions throughout the text. The title is misleading. It should be “Glucose-lowering and Beyond Glucose-lowering Effects of Imeglimin, a novel antidiabetic drug targeting mitochondrial bioenergetics and its effects on endothelial cells heart, kidney, and brain.” What did authors mean by having numbers 2, 3, and 5 in the title?

We changed the tile

From

Glucose-lowering and Beyond Glucose-lowering Effects of A Novel Anti-diabetic Drug with Action Targeting Mitochondrial Bioenergetics, Imeglimin: Its Effects on Endothelial Cells, Heart, Kidney and Brain

To

Glucose-lowering Effects of Imeglimin and Its Possible Beneficial Effects on Diabetic Complications

  1. According to the suggestion “The sentence starting with “an impact of anti-diabetic drugs on endothelial function etc.,” should be revised (L79).”

We deleted this sentence.

  1. According to the suggestion “The authors should include how the drug could improve pancreatic beta-cell function.”

We added the following sentences.

2.5. An improvement of -cell function

Imeglimin has a favorable effect on preservation of the number of insulin granules, recovery of morphology in mitochondria, and reduction of apoptosis [41]. How could imeglimin improve pancreatic -cell function? Reduced β-cell apoptosis may be due to decreased expression levels of various apoptosis- and/or inflammation-related factors such as inflammatory cytokines. A decrease of oxidative stress by imeglimin may also lead to reduce β-cell apoptotic cell death and to improve -cell function. There is the close association between the reduction of apoptotic β-cell death by imeglimin and the amelioration of β-cell function. When apoptotic β-cell death is induced, it is difficult for β-cells to preserve insulin biosynthesis and secretion. Although further studies should be performed, reduction of apoptotic β-cell death by imeglimin may improve -cell function. Briefly, the recovery of morphological change in β-cell mitochondria is likely to facilitate ATP production and thereby to enhance β-cell function. Furthermore, imeglimin-mediated improvement of structural integrity and homeostasis of ER may largely contribute to an improvement of β-cell function [34].

  1. According to the suggestion “How would the drug affect the glucose uptake by the skeletal muscles? “

We added the following sentences.

2.6. An enhancement of glucose uptake by the skeletal muscles

Skeletal muscle is the major tissue for insulin-mediated glucose disposal. De-creased uptake of glucose by muscle due to insulin resistance is an important factor for the development of type 2 diabetes. Acute administration of imeglimin significantly stimulated glucose uptake by muscle cells in a dose-dependent manner [36]. Muscle glucose uptake  was reduced in the streptozotocin-treated rats compared to the con-trol rats. Chronic administration of imeglimin (45 days) increased glucose uptake by such muscles. An improvement of mitochondrial dysfunction by imeglimin induces an improvement of insulin signaling in skeletal muscle [36]. An improvement of insulin resistance by imeglimin is expected to increase glucose transporter 4 (GLUT4) expres-sion and improve GLUT4 function in muscle, but this has not yet been proven.

  1. According to the literature, the drug inhibits hepatic glucose synthesis, not reduction, as mentioned by the authors.

We changed from “reduction” to “inhibition”.

  1. According to the suggestion “What are the side effects of the drug? Are there any records on that?”

We added the following sentences.

4.2. Reported treatment emergent adverse events (TEAEs)

According to the study evaluated long-term (52 weeks) safety of imeglimin, there were no serious drug-related TEAEs [48]. The TEAEs occurring in more than 5% of pa-tients with imeglimin monotherapy included nasopharyngitis, pharyngitis and nausea [48]. The TEAEs occurring in 3-5% of patients with imeglimin monotherapy included diarrhea, constipation, hypoglycemia and hyperglycemia [48].

Round 2

Reviewer 1 Report

The authors have addressed all the queries adequately therefore the revised version of manuscript is acceptable for publication.